# Breakdown Behavior of Metal Contact Positions in GaN HEMT with Nitrogen-Implanted Gate Using TCAD Simulation

**DOI:** 10.3390/mi13020169

**Published:** 2022-01-22

**Authors:** Gene Sheu, Yu-Lin Song, Ramyasri Mogarala, Dupati Susmitha, Kutagulla Issac

**Affiliations:** 1Department of Computer Science and Information Engineering, Asia University, Taichung 41354, Taiwan; g_sheu@asia.edu.tw (G.S.); ramya3620@gmail.com (R.M.); smithadupati@gmail.com (D.S.); kutagullaissac23@gmail.com (K.I.); 2Department of Bioinformatics and Medical Engineering, Asia University, Taichung 41354, Taiwan

**Keywords:** AlGaN/GaN, TCAD, HEMT, metal, 2DEG

## Abstract

In this study, the breakdown behavior of a calibrated depletion mode AlGaN/GaN transistor with a nitrogen-implanted gate region was simulated and analyzed using Sentaurus TCAD simulation, with particular emphasis on the metal contact design rule for a GaN-based high-electron-mobility transistor (HEMT) device with a variety of 2DEG concentrations grown on a silicon substrate. The breakdown behaviors for different source/drain contact schemes were investigated using Sentaurus simulation. The metal contact positions within the source and drain exhibited different piezoelectric effects and induced additional polarization charges for the 2DEG (two-dimensional electron gas). Due to the variation of source/drain contact schemes, electron density has changed the way to increase the electric field distribution, which in turn increased the breakdown voltage. The electric field distribution and 2DEG profiles were simulated to demonstrate that the piezoelectric effects at different metal contact positions considerably influence the breakdown voltage at different distances between drain metal contacts. When the contact position was far away from the AlGaN/GaN, the breakdown voltage of the nitrogen-implanted gated device decreased by 41% because of the relatively low electron density and weak induced piezoelectric effect. This reduction is significant for a 20 μm source-drain length. The minimum critical field used for the breakdown simulation was 4 MV/cm. The simulated AlGaN/GaN device exhibits different breakdown behaviors at different metal contact positions in the drain.

## 1. Introduction

Gallium nitride (GaN) is a promising material for next-generation power-switching devices. GaN is a group III–V material which retains the piezoelectric properties and spontaneous properties in nature [1]. GaN devices, such as high-electron-mobility transistors (HEMTs), metal insulator semiconductor HEMTs, and Schottky barrier diodes, have a high channel charge density (approximately 1 × $10^{13}\;{\mathrm{cm}}^{-2}$) at the interface between undoped GaN and AlGaN [2,3,4,5,6]. GaN has a wide bandgap and high mobility, heat capacity, breakdown field, saturation velocity, relative dielectric constant, and thermal conductivity [7,8,9,10,11,12]. The GaN-on-Si approach reduces the manufacturing cost of GaN power devices, because silicon substrates are low cost and have a large number of availabilities with high quality has attracted more and more of the industry’s attention [13]. However, GaN grown on Si exhibits some drawbacks, such as a high thermal expansion, defect density, and lattice mismatch. AlGaN/GaN devices have been considerably improved over the last decade, and the GaN-on-Si technology is currently widely used in industrial applications.

AlGaN/GaN HEMTs can be used in high-power-density and high-voltage operations because of their high carrier mobility in the two-dimensional electron gas (2DEG) region. These HEMTs have a high breakdown voltage because of their high critical field [14]. In power electronics applications, the breakdown voltage of high-voltage switching operations is selected with a margin to ensure stable operation. Present-day high-voltage AlGaN/GaN HEMTs are designed with a large breakdown voltage margin because their breakdown behaviors are complex on account of their many leakage paths and lack of avalanche withstanding capability [14]. A suitable breakdown voltage and the appropriate design rules for highly reliable AlGaN/GaN-on-Si devices have yet to be determined.

Because GaN can conduct heat, GaN devices can operate at high temperatures and in harsh environments [15,16,17]. The optimization of these GaN devices is still in its starting stages and has not discussed for the effect of piezoelectric polarization and spontaneous effect on the device performance.

To the best of our knowledge, this study is the first to investigate the breakdown mechanism of an AlGaN/GaN device with a nitrogen-implanted gate on a Si substrate. In addition, a Sentaurus simulation was conducted to propose a design rule for the metal contact position within the drain and determine the breakdown characteristics of the AlGaN/GaN device [18]. We attempted to identify a suitable metal contact position for ensuring a high breakdown margin and investigated the electric characteristics of the aforementioned device.

## 2. Simulation Device

The simulated AlGaN/GaN device contained a p-type Si <111> substrate. A Si substrate has a higher thermal conductivity than does a GaN substrate and is a suitable material for cost-effective processes requiring high thermal conductivity. The high thermal conductivity of a Si substrate allows it to be used to integrate high-power electronics. The GaN substrate of the simulated device was on a Si substrate with cracks and dislocations on its surface, which are caused by in-plane lattice constant mismatch [19]. This mismatch can be decreased using an aluminum nitride (AlN) layer. An epitaxial layer grown on an AlN nucleation layer reduces the tensile stress induced by a mismatch between Si and GaN during a cooling process [20,21]. The AlN layer protects the epitaxial layer from wafer bowing and wafer cracking. The epitaxial layer of the simulated device comprised a 3.9 µm-thick GaN/AlGaN/GaN/AlGaN buffer layer, on which an AlGaN layer and undoped GaN layer were implanted. The stacked buffer layer was doped uniformly with a carbon concentration of 1 × 10^18^ cm^−3^. This concentration implies that the GaN buffer can be grown under low-pressure settings at a residual carbon doping concentration [22]. On the undoped GaN layer, a 27 nm-thick AlGaN barrier layer was deposited, followed by a silicon nitride (Si_3_N_4_) passivation layer. At the interface between the undoped GaN layer and the AlGaN layer, a 2DEG (two-dimensional electron gas) channel region was formed. The 2DEG region shares two charges that are piezoelectrically polarized to induce strain and spontaneously polarized for the formation of bond electronegativity [23]. An Al mole fraction of approximately 22% was used in the AlGaN barrier layer to form the 2DEG. Si_3_N_4_ was deposited as the gate insulator through ohmic contact etching and metal deposition. The dimensions of the simulated device were as follows: gate length = 6 µm, source-to-gate distance = 3.5 µm, field plate length = 4.5 µm and gate-to-drain distance = 22.5 µm. The source and drain ohmic contact metal were annealed at 900 °C for 25 s. A gate contact was deposited over Si_3_N_4_ and followed by oxide as ILD (interlayer dielectric). On top, the silicon nitride passivation was deposited, and the device was achieved by nitrogen ion implantation. The schematic of the simulated AlGaN/GaN device is shown in Figure 1.

The device is 0 to 83 µm in the *x*-axis and 0 to 7.8 µm in *y*-axis. The nitrogen ion implantation profile was obtained by using the TRIM (transport of ions in matter) simulator [24], which calculates the interaction of matter. By using finite element Synopsys, Sentaurus software simulations were performed, and the creation of the device structure is by using the Sentaurus process, which is a process simulator equipped with the physical models that include default parameters calibrated with data from equipment vendors. The Sentaurus Process provides a predictive framework for simulating a broad range of technologies from nanoscale CMOS to large-scale high-voltage power devices. By using the Sentaurus device, the electrical characteristics were measured, which can simulates electrical behavior of a single semiconductor device in isolation or several physical devices combined in a circuit and supports the modeling of high mobility channel materials and implements highly efficient methods for modeling. Sentaurus Visual is used to visualize the output from the simulation.

Figure 2 illustrates the structures of the simulated device for different metal contact positions (marked with dotted lines). Figure 2a shows simulated device for metal contacts near channel region, Figure 2b shows simulated device for metal contacts in middle of source and drain regions, Figure 2c shows simulated device for metal contacts far away from channel region. Simulation physical models were selected from the TCAD simulation software. The models used for normally off AlGaN/GaN HEMT device simulation are listed in Table 1. The key parameters for tuning the device are donor-like traps −3 × 10^13^ cm^−2^ at the nitride/GaN interfaces. The control of the positive fixed charges −5 × 10^12^ cm^−2^, the acceptor traps activation energy in the AlGaN layer and buffer regions with 0.59 eV below the conduction band and the energy of the donor-like traps 1.42 eV below the conduction band were used [25].

A nitrogen ion implantation profile from TRIM simulator was simulated under an energy value of 300 keV and a dosage of 3 × 10^15^ cm^−2^. The 300 keV energy ensured that the 2DEG did not have a vacancy concentration of greater than 1 × 10^18^ cm^−3^. To convert a normally on device into a normally off device, the 2DEG region must be blocked to reduce the polarization charges. A nitrogen-implanted gate region with an adjustable dose can block the 2DEG region to ensure that the 2DEG density is at a suitable level for converting a normally on device to a normally off device. We found that an energy of 300 keV was appropriate for ensuring that the 2DEG concentration is less than 1 × 10^13^ cm^−3^. The total nitrogen vacancy concentration was approximately 1.3 × 10^17^ cm^−3^ (Figure 3).

## 3. Simulation Results and Discussion

Figure 4 displays the 2DEG density profile for metal contacts near the channel in the nitrogen-implanted gate region, to show the cut profile at 2DEG (interface of AlGaN and GaN undoped region).

The 2DEG density profiles for different metal contact positions are illustrated in Figure 5. The 2DEG region is defined from the quantum well with the thickness around 0.06 um to 0.09 µm. Piezoelectric properties affect the concentration and transport characteristics of 2DEG confined in the potential well at the GaN layers and the AlGaN layer interface and might lead to accumulation or depletion regions at the interfaces, depending on the polarity of the top surface. An increase in the 2DEG density caused by the piezoelectric effects is sometimes referred to as piezoelectric doping.

Figure 5 shows the piezoelectric effect of drain applied voltage (10 V, 30 V, 50 V……at breakdown point) which increased 2DEG charges. Figure 5a shows 2DEG density vs applied voltage for contacts near channel device, Figure 5b shows 2DEG density vs applied voltage for contacts at center, Figure 5c shows 2DEG density vs applied voltage for contacts far from channel device. An applied electric field at drain contact might influence the total piezoelectric effect. A piezoelectric AlGaN/GaN crystal is essentially an electromechanical transducer that changes strain into electrical potential and vice versa. Therefore, applying an external electrical field will cause some mechanical deformation, which may affect the total piezoelectric effect results in 2DEG density changes. The internal piezoelectric effect induced from the internal strain is caused by lattice mismatch between GaN layers and AlGaN layer.

Figure 6 depicts the simulated electric field distribution and 2DEG density of the AlGaN/GaN device. The dashed lines are drawn horizontally along the *x*-axis. The simulation results indicate that a stronger electric field corresponds to a higher 2DEG density induced by the piezoelectric effect caused by the voltage applied at a drain contact. The peak of 2DEG density and electrical field distribution increase respectively to the contact distance from the channel, indicating that more and more electrons are populated on the well. Combined with the property of high mobility, high-concentration electron gas in a potential well leads to 2DEG being the superior electrical conductivity. Based on this principle, 2DEG is widely used in field effect transistors. This ultrahigh switching ratio is attributed to the piezoelectric field which is perpendicular to the flowing direction of charge carriers and directly controls the opening and closure of the conducting channels in 2DEG, giving rise to excellent switching behavior [26].

Figure 7 shows the simulation results of electric field distribution and impact ionization under various metal contact conditions using the TCAD (technology computer-aided design) method. We found the electric field and impact ionization strengths are metal contact location dependent, and its magnitude decreases as the distance from the channel increases. The higher electric field strength results in higher breakdown voltage. We have also found from simulation that the higher electric field strength is caused by higher 2DEG density which is induced by the piezoelectric effect of applied voltage on the drain terminal. Both breakdown curves and the impact ionization hotspot map confirm that the breakdown mechanism is caused by impact ionization, but not surface breakdown. The minimum critical field used in simulation is 4 MV/cm. From previous studies, we know the temperature dependence coefficient of breakdown should be a positive value.

The depletion regions of GaN device structures with different metal contact positions are illustrated in Figure 8. The area which is in grey is depleted. The electric breakdown occurred at the drain side, and the marked region shows that for contacts near the channel, the device was completely depleted vertically along the *y*-axis at the drain side which shows in Figure 8a. Figure 8b contacts in the center device and Figure 8c contacts far from the channel device, sufficient space existed for depleting the device. This size of the depletion region was decreased by increasing the distance of the metal contact position from the channel. 

The transistor breakdown voltage was measured through TCAD device simulation. Figure 9 presents the variations in the transistor breakdown voltage with respect to the contact-channel separation. The transistor breakdown voltage was higher for metal contacts closer to the channel. As displayed in Figure 9, the simulated device had breakdown voltages of 75, 82.8, and 127.4 V when the metal contact was located far away from the channel, in the metal center and near the channel, respectively. The breakdown voltage increased by 41% for contacts near the channel device.

Figure 10 presents a plot of breakdown voltage against 2DEG density. We identified the metal contact position near the channel for achieving the maximum breakdown voltage for an uncapped AlGaN/GaN HEMT.

## 4. Conclusions

In this study, the breakdown behavior of high-voltage AlGaN/GaN HEMT with a nitrogen-implanted gate was analyzed by piezoelectric effects and two-dimensional device simulation using Sentaurus TCAD [27]. Impact ionization in the channel, which was triggered by electrons injected from the gate to the channel at a large reversed bias, was responsible for the breakdown [28]. The 2DEG density is found to increase for metal contacts in the drain region. The near-channel metal contact position increases more polarization charges induced by the piezoelectric effect than that of those away from the channel contact position. The distancing between metal contact and channel can affect the piezoelectric effect on 2DEG density by applied voltage at drain. Therefore, the piezoelectric-effect-induced polarization charges decrease as the distance of the contact from the channel increases. Consequently, the electric field and impact ionization should be suppressed to achieve a high breakdown voltage. The breakdown voltages for two contact positions within a 20 μm drain region under an operating voltage of 100 V can differ by as much as 41%. In conclusion, we formulated a new design rule governing the breakdown voltage of GaN HEMTs in this study.

## Figures and Tables

**Figure 1 micromachines-13-00169-f001:**
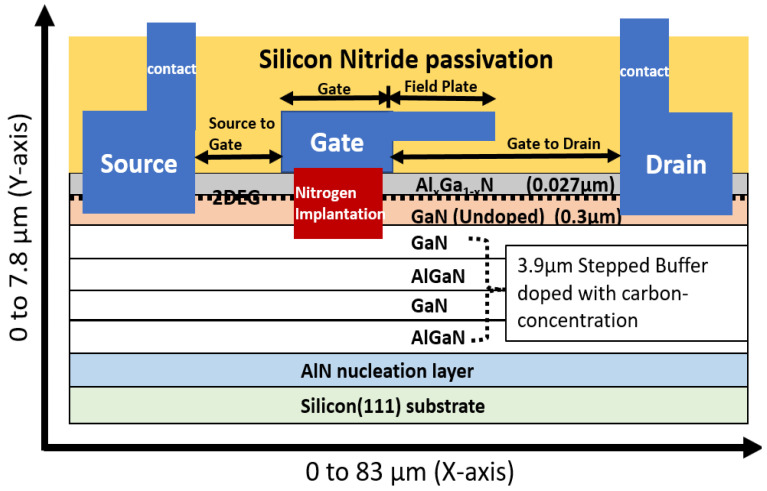
Schematic AlGaN/GaN HEMT device structure.

**Figure 2 micromachines-13-00169-f002:**
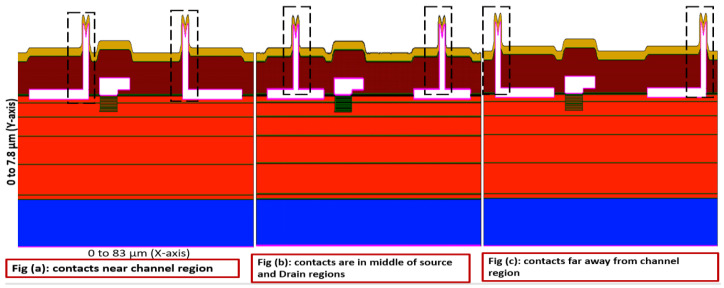
Simulated device structure with three various metal contact positions.

**Figure 3 micromachines-13-00169-f003:**
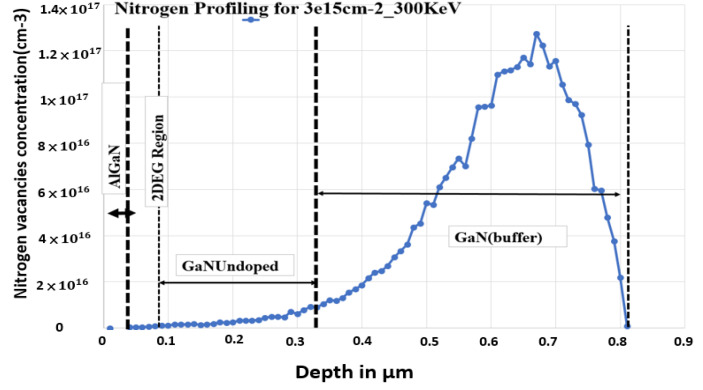
The depth profile of the nitrogen vacancies created by 300 KeV energy carried out by using TRIM simulation.

**Figure 4 micromachines-13-00169-f004:**
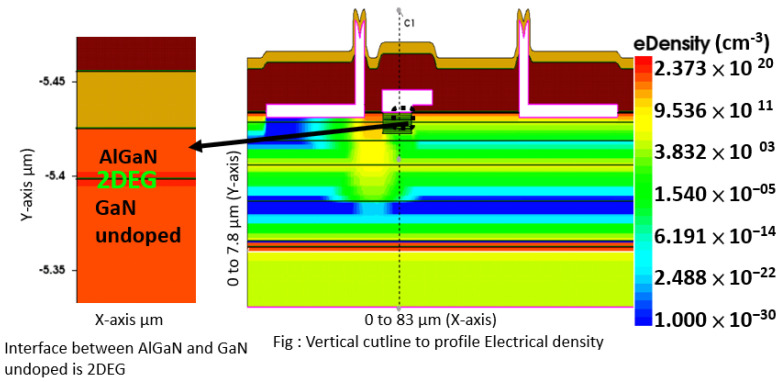
2DEG density device cut profile.

**Figure 5 micromachines-13-00169-f005:**
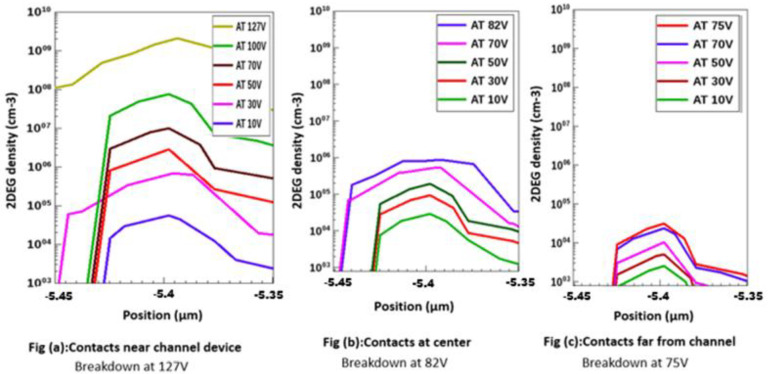
2DEG density vs. applied voltages with different metal contact positions.

**Figure 6 micromachines-13-00169-f006:**
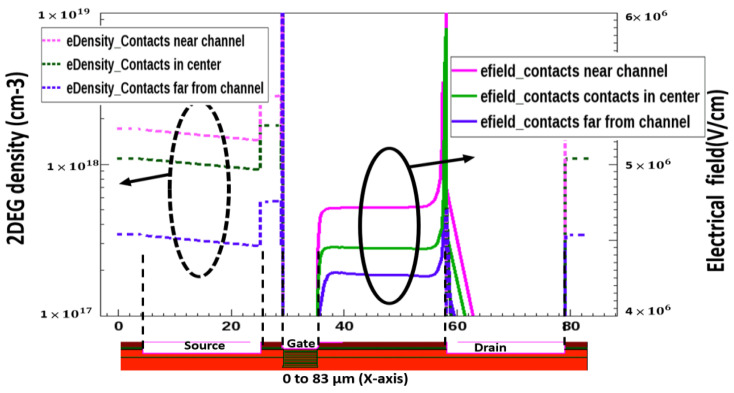
Simulation profile of 2DEG density and Electrical field distribution in AlGaN/GaN device.

**Figure 7 micromachines-13-00169-f007:**
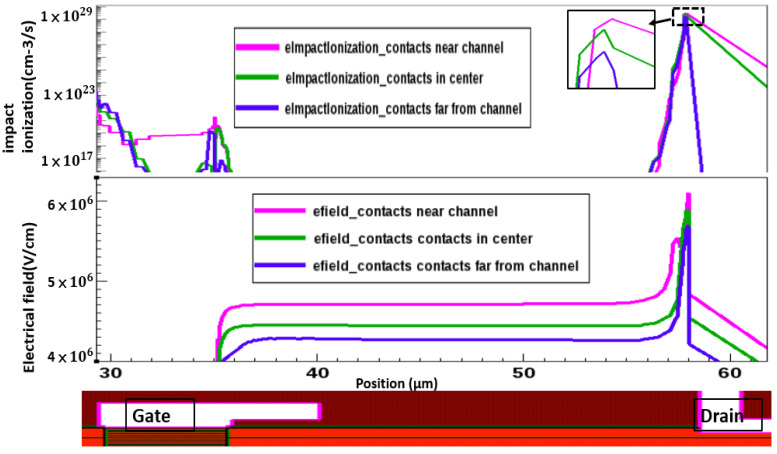
Impact ionization and electrical field profiles at breakdown for different metal contact positions.

**Figure 8 micromachines-13-00169-f008:**
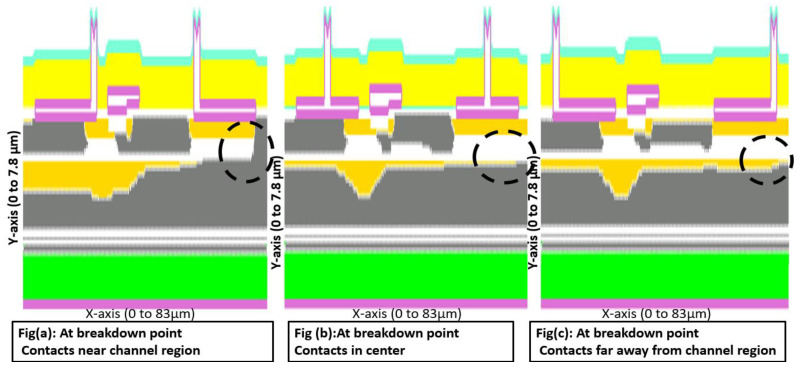
Depletion pattern with different metal positions.

**Figure 9 micromachines-13-00169-f009:**
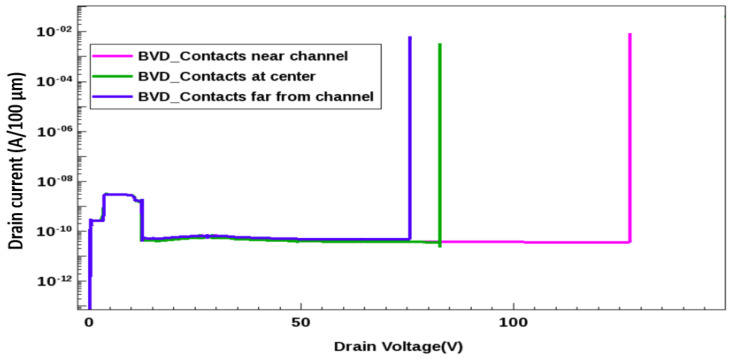
Simulated breakdown voltage comparison with different metal positions.

**Figure 10 micromachines-13-00169-f010:**
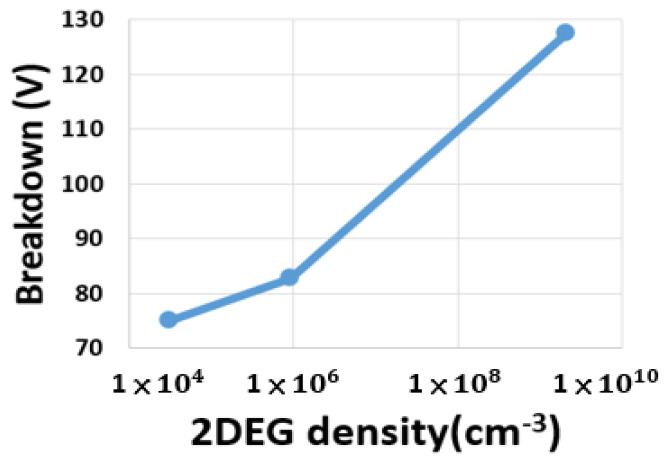
Breakdown vs. 2DEG density.

**Table 1 micromachines-13-00169-t001:** Physical models for simulation of normally off AlGaN/GaN HEMT device.

Physical Phenomenon	Model
Recombination	Shockley-Red-Hall
Mobility	High field saturation Doping dependence Poole frankel
Self-heating effect	Thermodynamic
Avalanche	Van overstraeten
Polarization	Piezo-Electric Strain Piezo-Electric Stress
Tunneling	Electron Barrier Tunneling

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
