# Peer review of "Breakdown Behavior of Metal Contact Positions in GaN HEMT with Nitrogen-Implanted Gate Using TCAD Simulation"

_micromachines, 2022, doi:10.3390/mi13020169_

Round 1

Reviewer 1 Report

First of all, I think it's a good simulation job via Sentaurus. But I miss some clear comparison with other works in the literature. It may be the first work where nitrogen implanted in the gate is used, but I think that the work would improve a lot if compared with values ​​obtained by other works with HEMT of GaN.

Figure 2 needs to be improved. The information of interest looks small. Perhaps eliminating layers that this figure does not have interest, the variation of the contacts could be better appreciated.

It seems that figure 4 is the result of the simulation of the structure shown in figure 2.a It would be convenient to comment on it to improve reading.

In figure 5 it is not explained why it is simulated in the range of voltages presented and why it is not carried out in all structures. It would also be interesting to explain what happens at 30V in the graph Fig 5.a where a change is seen with respect to close values.

In my opinion, it would be necessary to comment on the percentage of improvement in the magnitudes studied with respect to what exists in the state of the art. Perhaps a possible improvement is to add a comparative table with other works.

Reviewer 2 Report

This article shows interesting breakdown voltage results with respect to drain metal contact position on GaN HEMT structure from the TCAD simulations for device design rule application.  Few things need to be clarified.

  1. In line 84, “and gate-to-drain distance = 22.5µm” should be source-to-drain distance?
  2. Please label 2DEG in Figure 1. (The interface 2EDG region between AlGaN and undoped GaN was formed due to nitrogen implantation on gate.)
  3. If possible, please label the distance for the simulation device (HEMT) in Figure 1, gate length= 6µm, source-to-gate distance = 3.5µm and source-to-drain distance =22.5 µ At least to show the distance (no number) by arrows.
  4. How wide of 2DEG region (approx.) in your simulation?
  5. Need to mention the activities involved in your research in the abstract to march your title of article, such as using Sentaurus TCAD (technology computer aided design) software for simulation, GaN- based high-electron-mobility transistors (HEMT) devices and nitrogen implanted two-dimensional electron gas (2DEG) region etc.
  6. Keyword needs to add 2DEG.
